# In-hospital stress and patient outcomes: A systematic review and meta-analysis

Daniel M. Ford[1,2]*, Luke Budworth[3,4], Rebecca Lawton[1,2], Elizabeth A. Teale[5], Daryl B. O'Connor[1]

1 School of Psychology, Faculty of Medicine and Health, University of Leeds, Leeds, United Kingdom,
2 Quality and Safety Research Group, Bradford Institute for Health Research, Bradford Royal Infirmary, Bradford, United Kingdom, 3 NIHR Applied Research Collaboration Yorkshire and Humber, Bradford, United Kingdom, 4 Leeds Institute of Rheumatic and Musculoskeletal Medicine, University of Leeds, Leeds, United Kingdom, 5 Academic Unit for Aging and Stroke Research, University of Leeds, Leeds, United Kingdom

* psdmf@leeds.ac.uk

## Abstract

### Background

Hospital inpatients are exposed to high levels of stress during hospitalisation that may increase susceptibility to major adverse health events post-hospitalisation (known as post-hospital syndrome). However, the existing evidence base has not been reviewed and the magnitude of this relationship remains unknown. Therefore, the aim of the current systematic review and meta-analysis was to: 1) synthesise existing evidence and to determine the strength of the relationship between in-hospital stress and patient outcomes, and 2) determine if this relationship differs between (i) in-hospital vs post-hospital outcomes, and (ii) subjective vs objective outcome measures.

### Methods

A systematic search of MEDLINE, EMBASE, PsychINFO, CINAHL, and Web of Science from inception to February 2023 was conducted. Included studies reported a measure of perceived and appraised stress while in hospital, and at least one patient outcome. A random-effects model was generated to pool correlations (Pearson's $r$), followed by sub-group and sensitivity analyses. The study protocol was preregistered on PROSPERO (CRD42021237017).

### Results

A total of 10 studies, comprising 16 effects and 1,832 patients, satisfied the eligibility criteria and were included. A small-to-medium association was found: as in-hospital stress increased, patient outcomes deteriorated ($r = 0.19$; 95% CI: 0.12–0.26; $I^2 = 63.6$; $p < 0.001$). This association was significantly stronger for (i) in-hospital versus post-hospital outcomes, and (ii) subjective versus objective outcome measures. Sensitivity analyses indicated that our findings were robust.

**Data Availability Statement:** All relevant data are within the paper and its Supporting information files.

**Funding:** This research was funded by the National Institute for Health and Care Research (NIHR) Yorkshire and Humber Patient Safety Translational Research Centre (NIHR Yorkshire and Humber PSTRC) and the NIHR Yorkshire and Humber Applied Research Collaboration (NIHR Yorkshire and Humber ARC). The views expressed in this article are those of the authors and not necessarily those of the NIHR, or the Department of Health and Social Care.

**Competing interests:** The authors have declared that no competing interests exist.

## Conclusions

Higher levels of psychological stress experienced by hospital inpatients are associated with poorer patient outcomes. However, more high-quality, larger scale studies are required to better understand the association between in-hospital stressors and adverse outcomes.

## Introduction

Psychological stress is known to adversely influence health and wellbeing by causing negative changes in mental health outcomes and multiple physiological processes [1]. More specifically, stress has been shown to play a detrimental role in immune system dysfunction [2, 3], cardiovascular disease, coronary heart disease, and stroke [4]. In response to stressful encounters ('stressors'), the body veers from its homeostatic state, adjusting physiological parameters and releasing endocrinological mediators such as cortisol (the so-called "stress hormone"). This process of adapting is necessary for survival and is known as *allostasis* ("remaining stable by being variable" [see 5]). However, with prolonged exposure to stress, the body experiences excessive "wear and tear" from an inefficient management of stress mediators; a concept known as *allostatic load* [1, 6]. When this load becomes too great, the body experiences deleterious effects; a concept known as *allostatic overload* [7; see 8].

Allostatic overload is theorised to be the cause of *post-hospital syndrome* (PHS); an acquired period of generalised vulnerability to adverse events (e.g., post-operative wound infection) following hospitalisation [9]. Indeed, in some prominent conditions, only a third of all post-discharge readmissions (a proxy for poor post-hospital outcomes) were the same as that of the index admission [10]. This is even lower still for some conditions: the 30-day readmission for patients hospitalised due to acute myocardial infarction is approximately one in six [11], where only 10% of those readmissions were for a subsequent myocardial infarction [12]. Consequently, Krumholz [9] suggests that we should view the post-discharge period as a generalised syndrome of physiological impairment, rather than a routine recovery specific to the initial ailment.

More recently, the theorised, causal relationship between allostatic overload and PHS has been elaborated on by Goldwater and colleagues [13]. These authors have outlined several "hospital-related stressors" that are likely catalysts of allostatic overload: sleep disruption, malnourishment and dehydration, mobility restriction, and pain. However, this list is by-no-means comprehensive, there exists an unknown (and likely vast) number of these stressors, for example: loss of control [14], mental distress [15], equipment visibility [16], lack of light and nature [17], and, perhaps the most salient of all, relationships with staff [18, 19]. The combination of these stressors may make for an unpleasant experience for inpatients in their already vulnerable states [e.g., 20].

Indeed, it follows that, if stress causes deleterious effects, and if hospital stays expose patients to an assortment of stressors, then hospitalisation may be contributing to these adverse patient outcomes (this is the essence of PHS). Previous research has characterised hospitalisation as a traumatic event [e.g., 21, 22], even resembling interrogation [23], and has recorded that patient-reported hospital experiences are potentially associated with patient outcomes [22, 24–27]. In fact, regardless of stress, hospitalisation may be damaging for patients (particularly older adults), being a likely risk factor for cognitive decline [28, 29], functional decline [30–32], decompensated frailty [33], and new iatrogenic disability [34, 35].

Therefore, taken together, there is an immediate need for us to improve our understanding of in-hospital stress, and its effects on in-hospital and post-hospital patient outcomes. At present, the literature has not identified the strength of the relationship between in-hospital stress and patient outcomes. The current systematic review and meta-analysis will aim to do this by synthesising the existing evidence base of studies that have investigated the relationship between in-hospital stress–whereby stress is perceived and appraised by the patient during their hospital stay–and an in-hospital and/or post-hospital patient outcome.

### Research aims

The current review aims to synthesise the existing evidence base to determine the strength of the relationship between in-hospital stress and patient outcomes–broad definitions of these two variables are offered below. Secondary aims are to uncover whether the magnitude of this relationship differs between groups of outcomes: (i) in-hospital vs post-hospital, (ii) subjective (patient-reported) vs objective, and (iii) by study quality.

### In-hospital stress

O'Connor and Ferguson [36] describe three approaches that have been used in studying stress: the stimulus-based approach; the response-based approach; and the psychological interactional-appraisal approach. The latter is also known as the transactional model approach and has been defined as "a particular relationship between the person and the environment that is *appraised* by the person as taxing or exceeding his or her resources and endangering his or her well-being" [37 p. 19]. This appraisal is postulated to have two dimensions: a primary and secondary appraisal [37]. A primary appraisal evaluates the risks, demands, or challenges of a situation, while a secondary appraisal evaluates the availability of perceived resources and whether anything can be done to alter the outcome of the situation. Therefore, should two persons experience the same noxious event one person may appraise the situation as stressful (depending on the extent to which they perceive that they can meet its demands), while the other may not. Moreover, central to the transactional model approach is the notion that stress is a psychological construct that only arises when there is a mismatch between primary and secondary appraisal. Therefore, in the current review, in keeping with this approach, we will include any measure of stress that is perceived or appraised by a patient during their hospital stay.

### Patient outcomes

Outcomes following hospitalisation are varied; individual, specialty measures alone are not sufficient to gauge a patient's recovery. In their call for standardised patient outcomes, Porter and colleagues [38] postulate that patients are most concerned with the health status achieved, time, complications, suffering involved, and sustainability of benefits. For this reason, the current review will conduct a holistic approach to measuring hospital-related outcomes, under the umbrella term of *patient outcomes*. These outcomes will be sorted into two categories: subjective (e.g., self-rated, such as quality of life or pain) and objective (e.g., patient records, such as length of stay or readmission).

### Methods

The current review adhered to the PRISMA guidelines for the reporting of systematic reviews and meta-analyses [39].

## Eligibility criteria

Eligible studies were quantitative and included a measure of both: (i) in-hospital stress, whereby psychological stress was perceived and appraised by the patient during their hospital stay, and (ii) in-hospital and/or post-hospital patient outcome(s). Distress, measures of stress that did not include a perceived appraisal (e.g., cortisol levels), and studies focussing exclusively on participants with a psychiatric disorder (e.g., PTSD) were not included. Patient outcomes included clinical assessments, Patient-Reported Outcome Measures (PROMs; as defined by the Cochrane Handbook, Chapter 18 [40]), and patient records denoting quality of care (e.g., length of stay and readmission). Patient satisfaction was also included in the current review, as it has been included in previous systematic reviews measuring patient outcomes [e.g., 41–43], as well as the patient-reported outcomes chapter of the Cochrane Handbook, cited above. Routine in-hospital assessments (e.g., heart rate, body temperature, etc.), however, were not considered patient outcomes for the purpose of this review, as they are more likely to be markers of poor health, rather than an ailment in themselves. Similarly, Patient-Reported Experience Measure (PREM [e.g., 44]) were not included. Participants in the eligible studies were adults (18 years or older) that were hospital inpatients at the time in-hospital stress was measured. If the period spanned by the stress measure (e.g., "indicate how much each item has applied to you over the *past week*") covered more of the pre-hospital period than in-hospital period, then it was not included.

## Search strategy

Five databases were searched from inception to present: Medline, Embase, PsycINFO, CINAHL, and Web of Science. The search was first conducted on 5[th] July 2021 and updated on 2[nd] February 2023; and was limited by (i) English language, (ii) human studies, (iii) adults (18+ years), and (iv) peer-reviewed articles. All titles and abstracts were screened by the first author (D.F.), 20% of which were independently screened again (L.B.); any discrepancies were resolved via discussion. This process was repeated for full texts, with a third reviewer (D.OC or R.L) consulted where there was ambiguity or lack of agreement. Details of the protocol for this review were preregistered on PROSPERO (CRD42021237017), which can be accessed at https://www.crd.york.ac.uk/prospero/display_record.php?RecordID=237017.

## Search terms

The method of formulating search terms was adapted from the PICO framework [45] as shown in Table 1. Indexing terms were adapted as necessary for use in the databases searched (see Appendix A in S1 File for a full list of search terms for each database).

Outcome search terms were informed by several recently published systematic reviews measuring patient outcomes and using the same databases as the current review [e.g., 41–43]. These were amalgamated after the removal of unwanted terms: i.e., terms specific to these systematic reviews (e.g., "medication system errors") and those pertaining to routine in-hospital

**Table 1. PICO framework used to formulate search terms.**

| | |
|---|---|
| **Population:** | Adult inpatients |
| **Intervention (Exposure):** | In-hospital stress |
| **Comparison:** | Not applicable |
| **Outcome:** | Patient outcomes |

assessments (e.g., "blood pressure"). Post-hospital syndrome was considered a principal term to include as an outcome, and so was added to each search as a keyword.

## Data extraction

Data was extracted by D.F. and comprised: author, year, study design, recruitment method, country, sample size, age, sex, reason for treatment, length of stay, number of previous hospital stays, measure of in-hospital stress (including time frame of stress experienced, e.g., "in the past month"), and patient outcome measure (including length of follow-up). Where multiple patient outcome measures were present in a study, discussion between three of the reviewers (D.F., D.OC. R.L.) took place to determine which measure(s) was (were) most appropriate to include. For experimental studies, only control data was used. Where pre- and post-hospitali-sation patient outcome measures were recorded, post-measures were chosen as these were more representative of the hospital period. In-hospital patient outcomes measured at the same time as in-hospital stress were not included, as the nature of the causal relationship was unclear (e.g., pain measure in study by Volicer [46]).

## Quality assessment

The Effective Public Health Practice Project (EPHPP) quality assessment tool for quantitative studies was employed. This tool was chosen over others as it is more appropriate for observa-tional studies, while other options (e.g., Cochrane Risk of Bias tool) are more appropriate for randomised controlled trials. Each study was assessed on its design, method, and analysis, which informed an overall rating of the paper as "strong", "moderate", or "weak". Papers deemed as "weak" were not excluded from the overall analysis; rather, a subgroup analysis was conducted comparing the magnitude of association in these papers against those rated as "strong" or "moderate". All eligible studies were assessed by D.F. and L.B. using the chosen tool (See Appendix B in S1 File for individual assessment scores).

## Data analysis

Each study identified for inclusion in the review was inspected for research design, country, sample, stress measure, and patient outcome(s); these data were extracted and systematically recorded.

Meta-analyses were conducted using R Studio (version 4.1.3) [47] (all packages and code used are included in Appendix C in S1 File), employing random-effects modelling via the *metafor* package [48]. As we expected most of the eligible studies to employ a correlational design, we chose Pearson's *r* as the pooled effect size metric (using Fisher *z* to *r* back-transfor-mation method), where *r* = 0.10, 0.30, and 0.50 were considered small, medium, and large, respectively [49]. Unadjusted correlations were chosen over adjusted if both were provided in the paper. Where other statistics were reported, *r* was estimated using the Campbell Collabora-tion Effect Size Calculator [50].

Three sub-group analyses were planned a priori to address the secondary research ques-tions. Sub-groups were split by (i) strong and medium vs weak quality, (ii) in-hospital vs post-hospital outcomes, and (iii) subjective vs objective outcome measures. A meta-regression cal-culated whether the pooled effects of these sub-groups were significantly different. Meta-regression was also used to explore whether age and sex were significant covariates of the rela-tionship between in-hospital stress and patient outcomes.

Heterogeneity was assessed with Cochran's *Q* statistic and related $I^2$ statistic. Funnel plots were generated, and Egger's regression [51] was calculated to test for asymmetry, which assessed the risk of small study bias: an indicator of publication bias [52]. A selection model

[53] was also calculated to directly assess the risk of publication bias. All analyses were subject to leave-one-out sensitivity analyses [54] to observe how each study influenced the overall model. Any studies indicated as disproportionately influencing the model were excluded, with reason offered as to why the result of the study in question may be inaccurate.

## Results

Initial systematic searching yielded 2,227 records, plus three records identified through Google Scholar during the scoping review and feasibility stage, before the formal database search commenced. Following the PRISMA screening process guidelines [39], 10 studies remained for inclusion in the systematic review [46, 55–63], comprising 1,832 participants (Fig 1). All 10 studies were also suitable for meta-analysis; some studies did not record data for all variables we wished to extract (e.g., length of stay; see Table 2). There was 100% agreement between the two authors screening (D.F. and L.B.) on which studies to include and their quality assessment scores.

Studies were conducted in the following countries: four in the United States [46, 55, 58, 61], two in Australia [62, 63], one in Greece [59], one in India [57], one in Iran [56], and one in Turkey [60]. Studies were of varied design: four cross-sectional [55, 56, 59, 60], three cohort [46, 62, 63], one cohort analytic [58], one controlled clinical trial [57], and one randomised controlled trial [61]. All studies used convenience sampling, recruiting sample sizes of 91 to 535 across nine cohorts. However, it is important to note, only five of these 10 studies sought explicitly to address our research question [46, 55, 59–61]; while the other five studies assessed stress while in hospital, though this was not the main aim of the study.

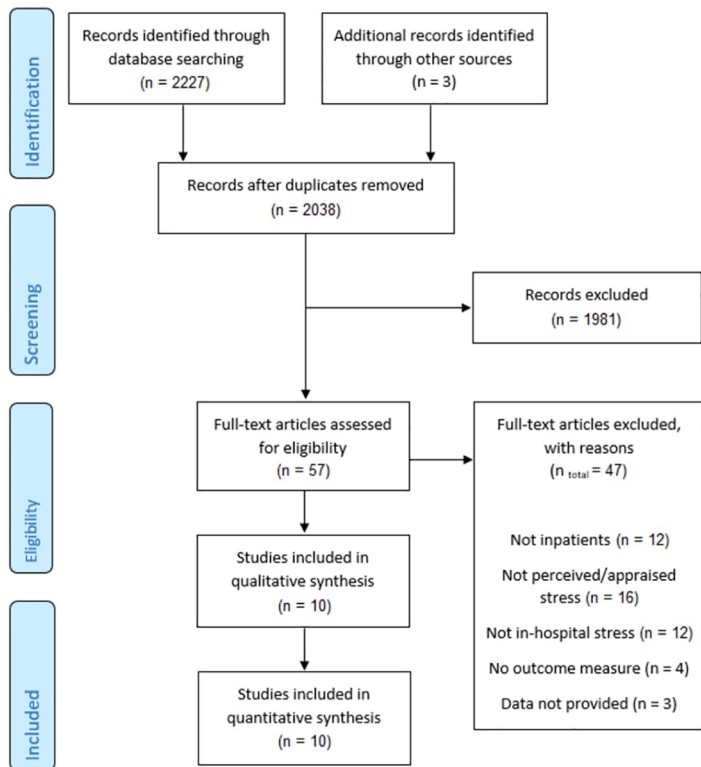

**Fig 1. PRISMA flow diagram presenting an overview of the selection process.**

**Table 2. Summary of studies included in the systematic review and meta-analysis.**

| Author, year (Country) | Sample | Mean Age (years) | Sex (% male) | Quality Score | Condition/ Treatment | Stress Measure | Outcome Measure (Subjective/Objective) | Summary Findings |
|---|---|---|---|---|---|---|---|---|
| Ahmadi, 1985 (United States) [55] | 100 | 58.3 | 42 | Weak | Mixed (Medical) | HSRS | Patient Satisfaction (Scale 1–10) **(S)**, Length of Stay **(O)** | Stress correlated with satisfaction ($r$ = -0.10, $p$ = n. s.) and length of stay ($r$ = 0.10, $p$ = n.s.). |
| Baharlooei et al., 2017 (Iran) [56] | 150 older adults | 68.0 | 48 | Weak | Type 2 Diabetes Complications (Medical) | DASS-21 | Length of Stay **(O)** | Stress correlated with length of stay ($r$ = 0.10, $p$ = n.s.). Multiple regression using stress as a predictor of length of stay (females: $\beta$ = 0.19, $p$ = 0.09; males: $\beta$ = 0.30, $p$ = 0.04). |
| Chalageri et al., 2021 (India) [57] | 45 (post-treatment control group only) | 34.5 | 91.1 | Weak | Spinal Cord Injury Rehabilitation (Medical) | PSS-14 | Quality of Life (WHOQOL-BREF) **(S)**, Spinal Cord Independence Measure (SCIM) **(O)**, Pain (NPRS) **(S)** | Stress correlated with quality of life* ($r$ = -0.84, $p < 0.001$), SCIM* ($r$ = -0.30, $p$ = 0.05), and pain* ($r$ = 0.36, $p$ = 0.02). |
| Edmondson et al., 2014 (United States) [58] | 225 | 62.5 | 66 | Weak | Acute Coronary Syndrome (Medical) | Telephone interview using a single item: "During the past two weeks, how often have you felt tense or 'wound up'?" | All-cause hospital readmission (within 30 days) **(O)** | Odds ratio* comparing readmission rates between high versus low stress groups ($OR$ = 2.39, $p$ = 0.11). |
| Karademas et al., 2009 (Greece) [59] | 128 | 58.0 | 55.5 | Weak | Mixed (Medical) | Three statements; scored on a five-point Likert scale | HRQOL **(S)**, Self-rated Health **(S)** | Stress correlated with HRQOL* ($r$ = -0.39, $p < 0.001$) and self-rated health ($r$ = -0.35, $p < 0.001$). |
| Karaer et al., 2021 (Turkey) [60] | 120 | 58.2 | 65.8 | Weak | Cardiovascular surgery (Surgical) | ICUESS | Satisfaction (ENCS) **(S)** | Stress correlated with satisfaction ($r$ = -0.38, $p < 0.001$). |
| Pati et al., 2016 (United States) [61] | 81 (control only) | 57.8 | 45.7 | Strong | Mixed (Medical & Surgical) | SACL | Length of Stay **(O)**, Pain **(S)** | Stress correlated with pain* ($r$ = 0.299, $p$ = 0.01), but not length of stay* ($r$ = 0.05, $p$ = 0.71). |
| Tully et al., 2008 (Australia) [62] | 222 | 63.1 | 83.2 | Moderate | Coronary Artery Bypass Grafting (Surgical) | DASS-42 | Unplanned, treatment-related Readmission (within six months) **(O)** | No significant difference in postoperative stress between readmitted and not readmitted patients ($p$ = 0.76). ** |
| Tully et al., 2011 (Australia) [63] | 226 | 63.1 | 83.2 | Moderate | Coronary Artery Bypass Grafting (Surgical) | DASS-42 | Incidence of Atrial Fibrillation **(O)** | No significant difference in postoperative stress between patients with and without incidence of postoperative atrial fibrillation ($p$ = 0.18). ** |
| Volicer, 1978 (United States) [46] | 535 | 52.0 | 41.7 | Moderate | Mixed (Medical & Surgical) | HSRS | Subjective physical status post hospital (POST) **(S)**, Return to usual activities (RETURN) **(S)** | Stress correlated (by averaging*** unadjusted correlations) with POST ($r$ = -0.21, $p < 0.001$) and RETURN scores ($r$ = -0.15, $p < 0.001$). |

*Figure calculated using data provided by the study (either reported in the paper or attained by emailing the author).

**Calculated via independent samples t-tests.

***See 'A note on averaging correlations' [73].

A variety of scales were used to measure stress while in hospital: three studies used the Depression, Anxiety and Stress Scale (DASS [64]), two studies used the Hospital Stress Rating Scale (HSRS [19]), one used the Perceived Stress Scale (PSS [65]), one used the Stress Arousal Checklist (SACL [66]), one used the Intensive Care Unit Environmental Stressor Scale (ICUESS [67]), one used a single-item interview question, and one used a three-item questionnaire. Within the 10 studies, there were 16 patient outcomes. Similarly, these measures were varied; three measured length of hospital stay, two measured satisfaction of care (rated 1–10 in study by Ahmadi [55]; ENCS [68]), two related to subjective health (rated 1–100 in study by Karademas [59]; Recovery Inventory [69]), two to quality of life (EQ-5D; [70]; WHOQOL-BREF [71]), two were self-rated pain measures (using numeric pain rating scales), two were incidence of readmission, one focussed on return to usual activities (rated 0–5 in study by Volicer [46]), one reported incidence of atrial fibrillation, and one used a spinal cord independence measure (SCIM [72]).

An unadjusted correlation was attained for each of the outcomes with their respective stress measures, with all 16 effects reporting in their predicted directions; nine of which reached statistical significance. As all effect sizes presented in their predicted directions (adverse outcomes correlated positively with stress; beneficial outcomes correlated negatively), it was possible to group both adverse and beneficial patient outcomes, temporarily ignoring the direction of the effect and focussing only on the magnitude. The random-effects model revealed a medium-sized, significant relationship between in-hospital stress and patient outcomes ($r = 0.27$; 95% confidence interval [CI], 0.12–0.41; n = 1,832; $p < 0.001$), with considerable heterogeneity ($I^2 = 92.7\%$, $p < 0.001$). However, one effect size was identified as an influential outlier, disproportionately influencing heterogeneity [74], and so was excluded from the remainder of the meta-analysis. The outlier was identified as quality of life in the study by Chalageri and colleagues [57], which was a near-perfect correlation ($r = 0.84$). We suspect that this is due to the two correlated measures quantifying similar constructs.

The remaining 15 correlations were suitable to be included in the full meta-analysis. The second random-effects model (Fig 2) revealed a small-to-medium, statistically significant relationship ($r = 0.19$; 95% CI: 0.12–0.26; n = 1,832; $p < 0.001$), with moderate heterogeneity ($I^2 = 63.6$, $p < 0.01$). As meta-analyses assume effect size independence, the use of robust variance estimation (RVE [75]) was necessary to account for within-subject statistical dependencies of studies that reported multiple outcomes. No notable differences were identified between the RVE model ($r = 0.19$; 95% CI: 0.09–0.30; $p < 0.01$) and unadjusted model (Fig 2), indicating that effect size dependencies were not disproportionately influencing the model.

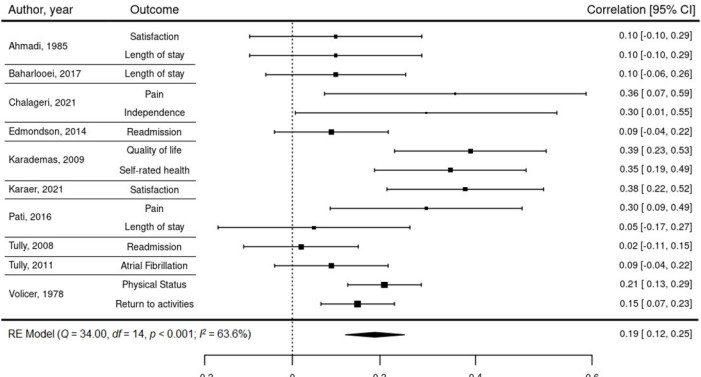

**Fig 2. In-hospital stress on patient outcomes: A forest plot of correlation coefficients within the included studies.**

## Sub-group analyses

Three pairs of models were produced, two of which addressed the secondary research questions, and the other observing the effect of study quality (Table 3). A statistically significant difference was reported for both of the relationships between in-hospital stress and (i) in-hospital versus post-hospital patient outcomes, and (ii) subjective verses objective outcome measures. In-hospital patient outcomes correlated more strongly with in-hospital stress than did those measured post-hospital ($QM = 4.23$, $p = 0.04$). Similarly, the effect was larger for subjectively measured patient outcomes than those measured objectively ($QM = 10.77$, $p < 0.001$). However, no significant difference was found in the effect sizes reported in strong and moderate studies versus weaker ones ($QM = 2.19$, $p = 0.14$).

A meta-regression was conducted to determine whether age and sex influenced the correlation between in-hospital stress and patient outcomes. Neither sex ($\beta = -0.0003$, $p = 0.86$) nor age ($\beta = -0.008$, $p = 0.08$) were identified as significant covariates. However, all but one of the studies reported mean age within a restricted range, between 52.0–68.0 years, and so this estimate may be inaccurate due to a lack of statistical power. It was not possible to test if length of stay (or other similar variables, such as number of previous hospital stays) was a significant covariate, as not enough studies reported this value, and some studies included length of stay as an outcome.

## Sensitivity analyses

The presence of publication bias was investigated. Egger's regression test was not statistically significant ($p = 0.176$), suggesting that there was no presence of small-study bias. However, a funnel plot of standard errors (Fig 3) showed that three studies may be missing to the left of the mean; this was supported by a Trim and Fill analysis [76], which shifted the x-intercept to the left by 0.041 (i.e., the pooled effect size decreased from: $r = 0.191$ to $r = 0.150$).

A selection model was calculated to directly address publication bias by giving more weight to effect sizes that were not statistically significant. A Likelihood Ratio test was then conducted, which indicated that the selection model ($r = 0.20$; 95% CI, 0.10–0.30; n = 1,832; $p < 0.001$) was not significantly different to the unadjusted model, suggesting that there was no evidence of publication bias ($X^2 = 0.054$, $p = 0.816$).

Systematically removing one of the 15 study correlations at a time, via leave-one-out analysis, indicated that no single effect size was disproportionately contributing to the model. The pooled effect size ranged from: $r = 0.173$ (-0.018) to $r = 0.205$ (+0.014), with each model remaining significant ($p < 0.001$).

## Discussion

The current review synthesised findings from 10 diverse studies that reported a measure of in-hospital stress and at least one patient outcome. A statistically significant association was identified between the two variables, consistent with previous systematic reviews observing the association between stress and health outcomes–including wound healing [77], cardiovascular disease [78], and poorer health outcomes generally [8, 79, 80]. However, the current systematic review is the first of its kind to look at patients' psychological stress specific to the in-hospital period; where the stressors are more numerous, and the body more vulnerable.

In these unadjusted analyses, a small-to-medium negative association was found, suggesting that as in-hospital stress increased, patient outcomes deteriorated, though no inference about causality can be made. The association was significantly stronger for subjective than objective outcome measures. This difference may be due to sources of information bias within the subjective measures, such as self-report bias and confirmation bias [81]. Indeed, these biases may also be compounded by common method variance [82]. Additionally, the observed differences

**Table 3. In-hospital stress on patient outcomes: Random-effects models of sub-groups.**

| Sub-groups of patient outcomes | Correlation ($0 < r < 1$) | 95% CI | Number of studies | $p$ value | Heterogeneity | |
|---|---|---|---|---|---|---|
| | | | | | $I^2$ | $p$ |
| **In-hospital** | 0.25 | [0.15; 0.34] | 10 | $< 0.001$ | 54% | 0.02 |
| **Post-hospital** | 0.13 | [0.06; 0.19] | 5 | $< 0.001$ | 46% | 0.13 |
| $QM = 4.23, p = 0.04$ | | | | | | |
| **Subjective** | 0.27 | [0.19; 0.36] | 8 | $< 0.001$ | 60% | 0.02 |
| **Objective** | 0.08 | [0.02; 0.14] | 7 | $< 0.01$ | 0% | 0.78 |
| $QM = 10.77, p < 0.001$ | | | | | | |
| **Strong/Moderate** | 0.14 | [0.07; 0.21] | 6 | $< 0.001$ | 46% | 0.09 |
| **Weak** | 0.24 | [0.14; 0.34] | 9 | $< 0.001$ | 61% | $< 0.01$ |
| $QM = 2.19, p = 0.14$ | | | | | | |

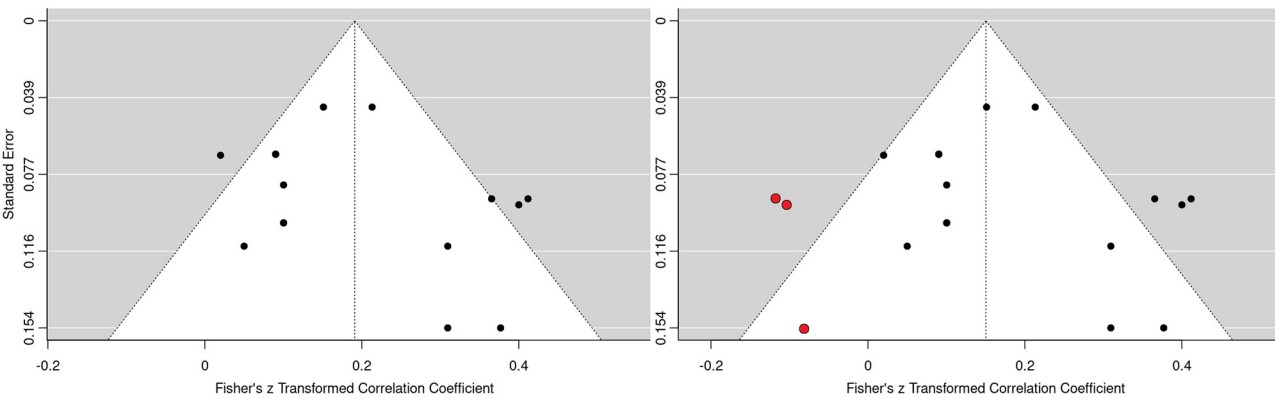

**Fig 3. Funnel plot (left) with trim and fill applied (right).**

are likely, in part, a result of the disparate nature of the two groups of measures. Subjective and objective measures in these studies tended to assess different types of outcomes; while subjective measures pertained to more complex and dynamic outcomes such as quality of life and subjective health, objective measures pertained more to outcomes associated with healthcare resource use such as length of stay and readmission. Nevertheless, the association between in-hospital stress and patient outcomes, albeit small, gives credence to Goldwater and colleagues' [13] theory that hospital-related allostatic overload may be a plausible aetiology of PHS [9].

Similarly, the association was significantly stronger for in-hospital patient outcomes than those measured post-hospital. Patients assessed in the post-hospital period are no longer exposed to in-hospital stressors, and so may not be experiencing the effects of PHS as acutely as their in-hospital counterparts as time has elapsed since the initial stressor exposure. However, other explanations for this difference in strength are the presence of case-mix (i.e., the differing types of patients treated) and the possibility that in-hospital stress is acting as a proxy for other associated and unmeasured confounding variables within the included studies. This may then be aggregated, again, by the disparate nature of the measures used to assess patient outcomes at the in-hospital versus post-hospital periods.

Meta-regression identified that neither sex nor age were statistically significant covariates; although, it is important to note that statistical power was too weak to draw any concrete conclusions. Other potential covariates, such as length of stay and number of previous hospital stays, were similarly not calculated on account of the limited number of studies. Previous literature would suggest that age is a significant covariate, where the association between stress and health increases with age. In their recent systematic review, Guidi and colleagues [8] outlined that allostatic overload, in older adults, is associated with frailty [83], cognitive and physical decline [e.g., 84–86], delirium [87], and risk of mortality [88]. Therefore, it is important that the role of age, in the context of in-hospital stress and patient outcome relations, is further investigated.

The results of the current systematic review and meta-analysis indicate that patient outcomes may be, in part, a function of the stress experienced by patients during their hospital stay. Should this relationship be investigated further, and causality is shown to be likely, emphasis should be placed on the need for (i) an increased focus on reducing the need for hospital admissions and (ii) greater attention to reducing the stress experienced by patients during their hospital stay. These actions must be culturally sensitive, and address healthcare at the individual and system levels [89]. If causation were to be established, reducing in-hospital

stress could be a cost-effective strategy for healthcare providers, given the association with longer stays and readmissions. The first logical step in this process would be to identify the specific aspects of hospitalisation that cause patients the most stress, such as the *hospital-related stressors* outlined by Goldwater and colleagues [13]. With this knowledge, appropriate policies and interventions can be implemented to reduce in-hospital stress, which may then lead to less adverse patient outcomes.

## Strengths and limitations

Our findings must be interpreted within the context of the limited academic literature. Consequently, the current review included relatively few articles, and reported on a variety of patient conditions, in-hospital stress measures, and patient outcome measures, which complicated attempts to make fair comparisons between studies. For example, incidence of atrial fibrillation (Tully et al., 2011) is an unusual outcome measure, and unlike any of the other outcomes included. Despite this, heterogeneity values were only moderate, leave-one-out analysis identified no statistical outliers, and every association within the included studies presented in their predicted directions–as in-hospital stress increased: beneficial patient outcomes (e.g., physical status, quality of life, etc.) deteriorated; and adverse patient outcomes (e.g., pain, readmission, etc.) increased.

Within the included studies, only half sought to address the research question of the current review, while the remaining articles were not specific to stress attributable to hospitalisation (though they did measure stress during the patients' hospital stays). Further, most of the included studies were deemed of weak quality, and only one was deemed strong. Most of the studies were cross-sectional and utilised either a correlational or non-randomised cohort design. Samples within the included studies were also limited in their ability to represent the wider population; all studies employed convenience sampling, most of which were limited to one-to-two wards within a single hospital. Evidently, more high-quality studies are essential to draw conclusions with sufficient confidence; these studies would ideally be large-scale, longitudinal, and randomised, using an agreed upon measure (e.g., HSRS) across multiple wards and hospitals.

Finally, despite only including studies where in-hospital stress was measured before the patient outcome, the presence of bidirectional relationships is entirely conceivable. For example, a patient may have had high levels of pain at the beginning of their hospital stay–at the time stress was measured–which would likely inflate the stress score, this pain would then be measured later, and assumed to be high due to an inflated stress score. In essence, in-hospital stress could be argued to be, at least in part, a proxy measure for a host of (undoubtedly stressful) factors that are antecedents to poorer patient outcomes, or even patient outcomes themselves. This ambiguity could have been partially accounted for by controlling for potential confounding variables (e.g., severity of illness), of which, few of the included studies measured. Alternatively, the design of more randomised controlled trials attempting to reduce in-hospital stress and measure patient outcomes [e.g., 61].

## Conclusion

This systematic review and meta-analysis found a small-to-medium relationship between in-hospital stress and a variety of patient outcomes. The association was stronger for in-hospital than post-hospital outcomes, and subjective than objective outcome measures. Our findings are comparable to other systematic reviews exploring the relationship between stress and health outcomes. Future research ought to aim to conduct high quality, large-scale studies (randomised, where possible) in order to make any conclusions with sufficient confidence.

These studies must account for confounding variables and employ a standardised measure of in-hospital stress.

## Supporting information

**S1 File.**
(DOCX)

**S2 File. PRISMA 2020 checklist.**
(DOCX)

## Author Contributions

**Conceptualization:** Daniel M. Ford, Rebecca Lawton, Daryl B. O'Connor.

**Data curation:** Daniel M. Ford, Luke Budworth.

**Formal analysis:** Daniel M. Ford.

**Investigation:** Daniel M. Ford.

**Methodology:** Daniel M. Ford.

**Resources:** Daniel M. Ford.

**Software:** Daniel M. Ford.

**Supervision:** Rebecca Lawton, Elizabeth A. Teale, Daryl B. O'Connor.

**Visualization:** Daniel M. Ford.

**Writing – original draft:** Daniel M. Ford.

**Writing – review & editing:** Luke Budworth, Rebecca Lawton, Elizabeth A. Teale, Daryl B. O'Connor.

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
