## [Decision Letter · Decision Letter 0]

25 Jan 2023

PONE-D-22-19657In-hospital stress and patient outcomes: A systematic review and meta-analysisPLOS ONE

Dear Dr. Ford,

Thank you for submitting your manuscript to PLOS ONE. After careful consideration, we feel that it has merit but does not fully meet PLOS ONE’s publication criteria as it currently stands. Therefore, we invite you to submit a revised version of the manuscript that addresses the points raised during the review process.

We look forward to receiving your revised manuscript.

Kind regards,

Walid Kamal Abdelbasset, Ph.D.

Academic Editor

PLOS ONE

Journal Requirements:

Reviewers' comments:

Reviewer's Responses to Questions

**Comments to the Author**

1. Is the manuscript technically sound, and do the data support the conclusions?

Reviewer #1: Yes

Reviewer #2: Yes

2. Has the statistical analysis been performed appropriately and rigorously? 

Reviewer #1: Yes

Reviewer #2: Yes

3. Have the authors made all data underlying the findings in their manuscript fully available?

Reviewer #1: No

Reviewer #2: Yes

4. Is the manuscript presented in an intelligible fashion and written in standard English?

Reviewer #1: Yes

Reviewer #2: Yes

5. Review Comments to the Author

Reviewer #1: Manuscript ID: PONE-D-22-19657

This is a systematic review and meta-analysis of the existing evidence between in-hospital stress and patient outcomes.

The topic of this manuscript is interesting. The paper is well written and straightforward.

I have some major concerns

1. The authors explained the concept of in-hospital stress and patient outcomes in the introduction, which I appreciated. However, I am not convinced about the relevancy to merge atrial fibrillation in the other outcomes. The atrial fibrillation was the only real medical complication that was included in these studies. Even I understand that comes from a study designed for atrial fibrillation, I found this outcome is really different from the others.

2. The authors said that the post-hospital syndrome was not a potential heading or a keyword for the databases. Did they require the assistance of a librarian? Although there is no specific heading, it should be possible to look for this concept in different databases. For example, in CINAHL, one could use (hospital-associated N3 (decline OR function* OR decondition* OR Deterioration))

OR (post-discharge N3 (decline OR function* OR decondition* OR cognit* OR Deterioration))

I am surprised that there was no studies with the post hospital syndrome.

Here are some minor concerns:

1. In the flow-chart, there was three records identified through other sources, please specify what they are.

2. In the first description table of patients, it would be useful to have some basic demographic data. For example: age, sex.

Reviewer #2: The authors conducted a systematic review and meta-analysis of 10 studies to determine the association between stress during hospitalization and outcomes. Although the review is based on a limited number of studies, the results are interesting because the psychological stress patients feel during their hospital stay is associated with adverse outcomes, especially in-hospital and subjective outcomes. In addition, the reviewer believes that the article presents a significant finding because such studies still need to be done more. The reviewer also finds the paper well-written, including the strengths and limitations of the study. A minor comment is that the studies included in the systematic review need to indicate which disease treatments.

6. PLOS authors have the option to publish the peer review history of their article (what does this mean?). If published, this will include your full peer review and any attached files.

Reviewer #1: No

Reviewer #2: No

---

## [Author Response · Author response to Decision Letter 0]

6 Feb 2023

6th February 2023

Dear Dr. Abdelbasset,

Manuscript ID: PONE-D-22-19657

In-hospital stress and patient outcomes: A systematic review and meta-analysis

Thank you for providing us with detailed feedback for our paper. We have addressed each of the points raised by Reviewers #1 and #2 and believe that this has added further clarification within the method and results sections. We hope that you feel we have satisfactorily responded to the points raised or have provided sufficient justification where an alternative approach has been taken. We feel that the manuscript is now suitable for publication in PLOS ONE. However, we will be very happy to respond to any additional comments.

For your reference, we have addressed each of the reviewers’ comments below, and have provided detail on where and how we have addressed each comment.

Reviewer #1:

This is a systematic review and meta-analysis of the existing evidence between in-hospital stress and patient outcomes.

The topic of this manuscript is interesting. The paper is well written and straightforward.

I have some major concerns

1. The authors explained the concept of in-hospital stress and patient outcomes in the introduction, which I appreciated. However, I am not convinced about the relevancy to merge atrial fibrillation in the other outcomes. The atrial fibrillation was the only real medical complication that was included in these studies. Even I understand that comes from a study designed for atrial fibrillation, I found this outcome is really different from the others.

Response: We agree with the reviewer and appreciate that incidence of atrial fibrillation is different to the other outcome measures. As a result, we have added a comment on page 21 that acknowledges this issue. Moreover, it is worth noting that this study met the review inclusion criteria and therefore we think it appropriate to leave in the meta-analysis. However, we have conducted a leave-one-out analysis (see page 19) to ensure that inclusion of this study (or any other study) is not having a disproportionate influence on our results. To this end, we have stated that:

“Systematically removing one of the 15 correlations at a time, via leave-one-out analysis, indicated that no single effect size was disproportionately contributing to the model. The pooled effect size ranged from: r = 0.173 (-0.018) to r = 0.205 (+0.014), with each model remaining significant (p < 0.001)”.

2. The authors said that the post-hospital syndrome was not a potential heading or a keyword for the databases. Did they require the assistance of a librarian? Although there is no specific heading, it should be possible to look for this concept in different databases. For example, in CINAHL, one could use (hospital-associated N3 (decline OR function* OR decondition* OR Deterioration))

OR (post-discharge N3 (decline OR function* OR decondition* OR cognit* OR Deterioration))

I am surprised that there was no studies with the post hospital syndrome.

Response: We have re-run the searches including ‘post-hospital syndrome’ and ‘posthospital syndrome’ as keywords (see S1_Appendices.docx). This also allowed us to update the search from July 2022 to February 2023. This yielded 113 further papers to screen (after de-duplication), but no additional studies were eligible for inclusion in the systematic review and meta-analysis. We have updated search result numbers in-text (pages 2 & 11) and within the PRISMA flow diagram (page 12). We have also updated the search term justification (page 8) to include the new keyword.

The reason for this absence of additional studies is due to the post-hospital syndrome (PHS) literature typically measuring PHS exposure as a hospital admission [30 days / 90 days / one year] prior to the current hospital admission (e.g. Hart et al., 2020; Kirshenbaum et al., 2019; Lim et al., 2020; Sharoky et al., 2017). In our review, we wished to look more specifically at the level of stress experienced in hospital, rather than the mere presence of a hospital admission (which would be a different review entirely).

Additionally, we thank the reviewer for their elegant solution to include the effects of post-hospital syndrome in the search without the specific heading. However, re-running the CINAHL search with the two examples yielded no additional papers.

Hart, S. T., Nelson, M., Kirshenbaum, E., Chen, Y., Mueller, E. R., & Gupta, G. (2020). Post-hospital syndrome predicts poor postoperative outcomes and increased cost following transvaginal midurethral sling placement. International Urogynecology Journal, 31, 1417-1422.

Kirshenbaum, E. J., Nelson, M., Hehemann, M. C., Kothari, A. N., Eguia, E., Farooq, A., ... & Santos, G. D. (2019). Impact of post-hospital syndrome on penile prosthesis outcomes: a period of global health risk. The Journal of Urology, 201(1), 154-159.

Lim, S., Alarhayem, A. Q., Farivar, B., Smolock, C. J., Kirksey, L., Caputo, F. J., ... & Hardy, D. M. (2020). Impact of posthospital syndrome on outcomes of elective endovascular repair of abdominal aortic aneurysm. Journal of Vascular Surgery, 72(5), 1618-1625.

Sharoky, C. E., Collier, K. T., Wirtalla, C. J., Sinnamon, A. J., Neuwirth, M. G., Kuo, L. E., ... & Kelz, R. R. (2017). Hospitalization in the year preceding major oncologic surgery increases risk for adverse postoperative events. Annals of Surgical Oncology, 24, 3477-3485.

Here are some minor concerns:

1. In the flow-chart, there was three records identified through other sources, please specify what they are.

Response: the three articles were identified in Google Scholar during the scoping review and feasibility stage before the formal database search commenced. This has now been clarified on page 11.

2. In the first description table of patients, it would be useful to have some basic demographic data. For example: age, sex.

Response: participants’ age and sex have been added to Table 2 (pages 13-15).

Reviewer #2:

The authors conducted a systematic review and meta-analysis of 10 studies to determine the association between stress during hospitalization and outcomes. Although the review is based on a limited number of studies, the results are interesting because the psychological stress patients feel during their hospital stay is associated with adverse outcomes, especially in-hospital and subjective outcomes. In addition, the reviewer believes that the article presents a significant finding because such studies still need to be done more. The reviewer also finds the paper well-written, including the strengths and limitations of the study. A minor comment is that the studies included in the systematic review need to indicate which disease treatments.

Response: participants’ condition/treatment have been added to Table 2 (pages 13-15).

---

## [Decision Letter · Decision Letter 1]

23 Feb 2023

In-hospital stress and patient outcomes: A systematic review and meta-analysis

PONE-D-22-19657R1

Dear Dr. Ford,

We’re pleased to inform you that your manuscript has been judged scientifically suitable for publication and will be formally accepted for publication once it meets all outstanding technical requirements.

Kind regards,

Walid Kamal Abdelbasset, Ph.D.

Academic Editor

PLOS ONE

Additional Editor Comments (optional):

Reviewers' comments:

Reviewer's Responses to Questions

**Comments to the Author**

1. If the authors have adequately addressed your comments raised in a previous round of review and you feel that this manuscript is now acceptable for publication, you may indicate that here to bypass the “Comments to the Author” section, enter your conflict of interest statement in the “Confidential to Editor” section, and submit your "Accept" recommendation.

Reviewer #1: All comments have been addressed

Reviewer #2: All comments have been addressed

2. Is the manuscript technically sound, and do the data support the conclusions?

Reviewer #1: Yes

Reviewer #2: (No Response)

3. Has the statistical analysis been performed appropriately and rigorously? 

Reviewer #1: Yes

Reviewer #2: (No Response)

4. Have the authors made all data underlying the findings in their manuscript fully available?

Reviewer #1: Yes

Reviewer #2: (No Response)

5. Is the manuscript presented in an intelligible fashion and written in standard English?

Reviewer #1: Yes

Reviewer #2: (No Response)

6. Review Comments to the Author

Reviewer #1: Dear authors,

Thank you for this revised version. I feel my comments have been fairly addressed.

I don't have further comments.

Reviewer #2: The authors revised the manuscript for each comment that I have pointed out, and the revised version is satisfactory. Therefore, I think that the revised version is now suitable for publication.

7. PLOS authors have the option to publish the peer review history of their article (what does this mean?). If published, this will include your full peer review and any attached files.

Reviewer #1: **Yes: **Yaohua CHEN

Reviewer #2: No

---

## [Editor Report · Acceptance letter]

27 Feb 2023

PONE-D-22-19657R1 

In-hospital stress and patient outcomes: A systematic review and meta-analysis 

Dear Dr. Ford:

I'm pleased to inform you that your manuscript has been deemed suitable for publication in PLOS ONE. Congratulations! Your manuscript is now with our production department. 

Kind regards, 

on behalf of

Dr. Walid Kamal Abdelbasset 

Academic Editor

PLOS ONE